# Selective Laser Sintering of High-Temperature Thermoset Polymer

Md Sahid Hassan [1,2,*], Kazi Md Masum Billah [3], Samuel Ernesto Hall [1,2], Sergio Sepulveda [1,2], Jaime Eduardo Regis [1,2], Cory Marquez [1,2], Sergio Cordova [1], Jasmine Whitaker [4], Thomas Robison [4], James Keating [5], Evgeny Shafirovich [1] and Yirong Lin [1,2,*]

1 Department of Mechanical Engineering, The University of Texas at El Paso, El Paso, TX 79968, USA; sehallsanchez@miners.utep.edu (S.E.H.); sdsepulveda2@miners.utep.edu (S.S.); jeregis@miners.utep.edu (J.E.R.); cmarquez10@miners.utep.edu (C.M.); scordova4@miners.utep.edu (S.C.); eshafirovich2@utep.edu (E.S.)
2 W.M. Keck Center for 3D Innovation, The University of Texas at El Paso, El Paso, TX 79968, USA
3 Mechanical Engineering Program, University of Houston-Clear Lake, Houston, TX 77058, USA; billah@uhcl.edu
4 Honeywell FM&T, Kansas City, MO 64147, USA; jwhitaker@kcnsc.doe.gov (J.W.); trobison@kcnsc.doe.gov (T.R.)
5 Imitec Inc., Schenectady, New York, NY 12308, USA; james.keating@imitec.com
* Correspondence: mhassan2@miners.utep.edu (M.S.H.); ylin3@utep.edu (Y.L.)

**Abstract:** Thermoplastic materials such as PA12 and PA6 have been extensively employed in Selective Laser Sintering (SLS) 3D printing applications due to their printability, processability, and crystalline structure. However, thermoplastic-based materials lack polymer inter-chain bonding, resulting in inferior mechanical and thermal properties and relatively low fatigue behavior. Therefore, 3D printing of high-performance crosslinked thermosets using SLS technology is paramount to pursue as an alternative to thermoplastics. In this work, a thermoset resin was successfully 3D printed using SLS, and its thermal stability of printed parts after a multi-step post-curing process was investigated. Dimensionally stable and high glass transition temperature ($T_g$: ~300 °C) thermoset parts were fabricated using SLS. The polymer crosslinking mechanism during the printing and curing process was investigated through FTIR spectra, while the mechanical stability of the SLS 3D-printed thermoset was characterized through compression tests. It is found that 100% crosslinked thermoset can be 3D printed with 900% higher compressive strength than printed green parts.

**Keywords:** selective laser sintering (SLS); printability; thermoset; thermal stability; glass transition temperature; polymer crosslinking; mechanical stability; compressive strength

## 1. Introduction

Selective Laser Sintering (SLS) is an additive manufacturing (AM) technique (i.e., 3D printing) that mainly deals with polymer materials in the powder form, which has the advantages of manufacturing complex objects and shapes without the need of support structures, compared to other AM techniques [1]. The core principle of SLS technology is to fuse the particles using a laser beam that selectively scans the powder bed, then deposits new layers of powder and repeats until the geometry is complete. Specific parameters should be considered during the sintering process design for every material in powder form, such as laser energy density, part bed temperature, layer thickness, hatch distance, laser power, and optimum wavelength for energy absorption [2–4]. Additionally, the powder's morphology and granulometry are well-known to be crucial parameters in the SLS method [5].

A significant amount of effort has been made to study diverse materials to use as a feedstock to produce parts with improved mechanical properties for specific applications.

For example, 3D printing ceramics by SLS for sophisticated and well-defined structures such as gears and casting molds were fabricated using Aluminum Oxide ($Al_2O_3$)-based mixed powder. Such complex shapes were manufactured by adding different binders in the matrix that enhanced the material's properties [6,7]. Additionally, they carry the practicability to fabricate complex shapes as functional parts, such as the Bionic Handling assistant using PA-12, a common thermoplastic material that shows good processability for SLS [8]. In addition, the applications of SLS depend upon the requirements of the material properties for the desired application, such as fabrication of biomedical devices, which demands more remarkable accuracy and control over producing intricate geometric figures [9], or even the potential of printing materials that can withstand hot temperatures for aerospace applications.

Regarding the 3D printing of thermoset powders, relatively less known information can be gathered to develop the processing parameters due to the complex curing mechanism than the widely used thermoplastic powder [10]. The focus on these polymer materials relies on increasing the application temperature, overall strength, and stability when reaching well-defined parameters for printing with a specified material. Notably, a research effort was made by the NASA Glenn Research Center to investigate the possibility of SLS 3D printing of a melt-processable RTM370 imide resin powder terminated with reactive phenyl ethynyl groups, following post-curing to achieve crosslinking and higher-temperature applications [11]. The main challenge was defining optimized processing parameters to ensure dimensionally stable parts with improved mechanical properties. In addition, crosslinking after the post-curing stage was difficult due to the low molecular weight and viscosity feedstock [12]. However, this research showed a path to the 3D printing of complex thermoset resin that needs additional and successful demonstrations to be adopted in the applications above 300 °C.

Additionally, SLS 3D printing of thermoset-based epoxy resin was performed by implementing alternative strategies by implementing the alternative strategy of targeting timing-based pre-curing of the resin that was employed to minimize the additional curing [13]. This allowed a sub-Tg cure profile to be employed to achieve initial gelation and thus allowed the material to be fully cured above its $T_g$ with minimal sagging.

Therefore, focusing on thermoset materials, this work reports the SLS 3D printing of thermoset bismaleimide (BMI) resin to demonstrate its printability and thermal stability. This paper shows the optimization of SLS printing parameters for BMI thermoset powder to obtain dimensionally accurate printed parts, along with its successful curing process. In addition, it quantifies the degree of curing at different curing stages by Fourier Transform Infrared Spectroscopy (FTIR) and Differential Scanning Calorimetry (DSC), and proves a significant improvement in mechanical properties after post-curing treatment. Additionally, this research demonstrates the printing and curing of Dynamic Mechanical Analysis (DMA) samples and performs tests to analyze the thermal stability by improving the glass transition temperature ($T_g$) point and exploring the damping factor.

## 2. Experimental Methodology

### 2.1. Materials

Bismaleimide (BMI I-841) powder was used as a thermoset material. BMI I-841 powder was produced, further staged, and advanced (molded at a higher temperature for an extra two hours to reduce the gelation time) by Imitec Inc. (New York, NY, USA) to make it printable for SLS printing. Carbon Micro-Balloons (CMB) (Honeywell, NJ, USA) with 5–30 micron diameters and 2.3 micron wall thickness were used as filler materials.

### 2.2. Synthesis of Composite BMI/CMB Powder

For successful SLS printing, the particle size should be 20 to 80 μm and spherical in shape [14]. Spherical particles ensure the flowability and homogenous distribution of powder particles during the SLS process. The Gilson SS-15D sieve shaker machine was used to sieve the powder and to obtain a standard particle size distribution in the range of

20 to 80 μm. Virgin BMI powder was used to print at first; however, due to the powder's bright yellow color, the laser from the diode laser source was reflected instead of being absorbed by the powder [15] during the printing process. Therefore, CMBs (10 vol%) were blended with sieved BMI powder to increase its laser absorbability. A glass test tube was used to measure the volume of each of the powders. Initially, the mixing of the powders was performed manually. Later, the Bioengineering Inversina mixture machine was used to mix the two powders at 60 rpm and at room temperature for 2 h to obtain homogeneous powder blends.

### 2.3. Specimen Fabrication Using the Selective Laser Sintering Process

The SLS 3D printing was performed using the Sinterit Lisa 3D printer (3D Herndon, Herndon, VA, USA). The machine was equipped with a 5-Watt Diode laser. Advanced and upgraded 'Sinterit Studio' software was used to develop the printing parameters for the BMI/CMB blended powder. 3D printing temperatures, laser power, laser movements, and geometries at different stages of the printing process were adjusted through this software. The bed temperature of the SLS 3D printer was guided by performing Differential Scanning Calorimetry (DSC) characterization on BMI powders. The DSC result was used to understand BMI's melting point, which ultimately helped to set the bed temperature at 105 °C (Section 3.4).

Initially, multiple samples were fabricated at different printing parameters. Visual inspection was performed to assess each part's print quality, as shown in the Results and Discussion Section. Based on the evidence of the dimensional stability, part density, and visual inspection, the printing parameter window was developed and finalized (as shown in Table 1).

**Table 1.** Optimized SLS process parameters for BMI/CMB thermoset powder.

| Process Parameter | Value |
|---|---|
| Laser Power | 5 W |
| Layer height (LH) | 0.150 mm |
| Hatch spacing (HS) | 0.36 mm |
| Full layer feed ratio (FLFR) | 2.0 |
| Energy scale (ES) | 1.0 |
| Max energy per (MEP) $cm^3$, infill | 700 |
| Constant energy (CE), infill | 0.8 |
| Max power depth (MPD), infill | 2.5 |
| Max energy per (MEP) $cm^3$, perimeters | 700 |
| Constant energy (CE), perimeters | 0.8 |
| Max power depth (MPD), perimeters | 2.5 |

According to the printer manufacturer's guidelines [16], maximum energy per unit volume ($cm^3$) (MEP) and constant energy (CE) are the two most effective laser power parameters which directly impact the final energy density. MEP and CE are directly dependent on the Maximum Power Depth (MPD) value. MEP has a small impact on laser energy through the first layers with a notable effect on layers at depths equal to or higher than the defined MPD value. On the other hand, CE has a high impact on laser energy through the first layers but a less significant effect on the layers at depths equal to or higher than the defined MPD. As the impact of MEP on final energy density gradually increases from the first layer of the specimen to the defined MPD value, the scan speed of the printer decreases stepwise so that the absorbed laser energy can be developed as the printing proceeds, and after the defined MPD value, the energy density will remain constant up to the last layer of the specimen. For BMI/CMB powder, with the optimized

printing parameter set, the average scan speed was 12.16 mm/s, which was recorded by a high-resolution camera. Average energy density can be defined as,

$$\text{Average Energy Density} \; = \frac{P}{V \times LH \times SH} \tag{1}$$

where P = laser power—5 W for the Sinterit Lisa SLS printer.

　　V = average scan speed, 12.16 mm/s.

　　LH = layer height, 0.150 mm.

　　HS = hatch spacing, 0.36 mm.

　　Therefore, using the above-mentioned printing parameters, the average energy density was 7.6 J/mm$^3$ and it was used for printing the BMI/CMB powder blend. The Sinterit sandblaster (3D Herndon, Herndon, VA, USA) was used for de-powdering of fabricated green parts, and dimensions were measured by a vernier caliper.

### 2.4. Curing of SLS 3D-Printed Specimen

　　In order to ensure the stability and crosslinking in the molecular structure in the SLS printed BMI/CMB green body, curing was accomplished using a forced convection oven (Lab Companion, Jeio Tech, Daejeon, Korea). As suggested by the BMI manufacturer, 3D-printed parts were cured at 250 °C for 4 h to fully crosslink the polymer chains. However, for the complex geometry part, it could not hold its shape by following the same curing process at a single step due to the curing temperature exceeding T$_g$ of polymer right after printing. Hence, multiple temperature steps were followed for curing each sample. Figure 1 shows the step-by-step curing process of an SLS 3D-printed complex specimen at elevated temperatures. The 3D printer complex demonstration model was downloaded as an STL file from GrabCAD [17].

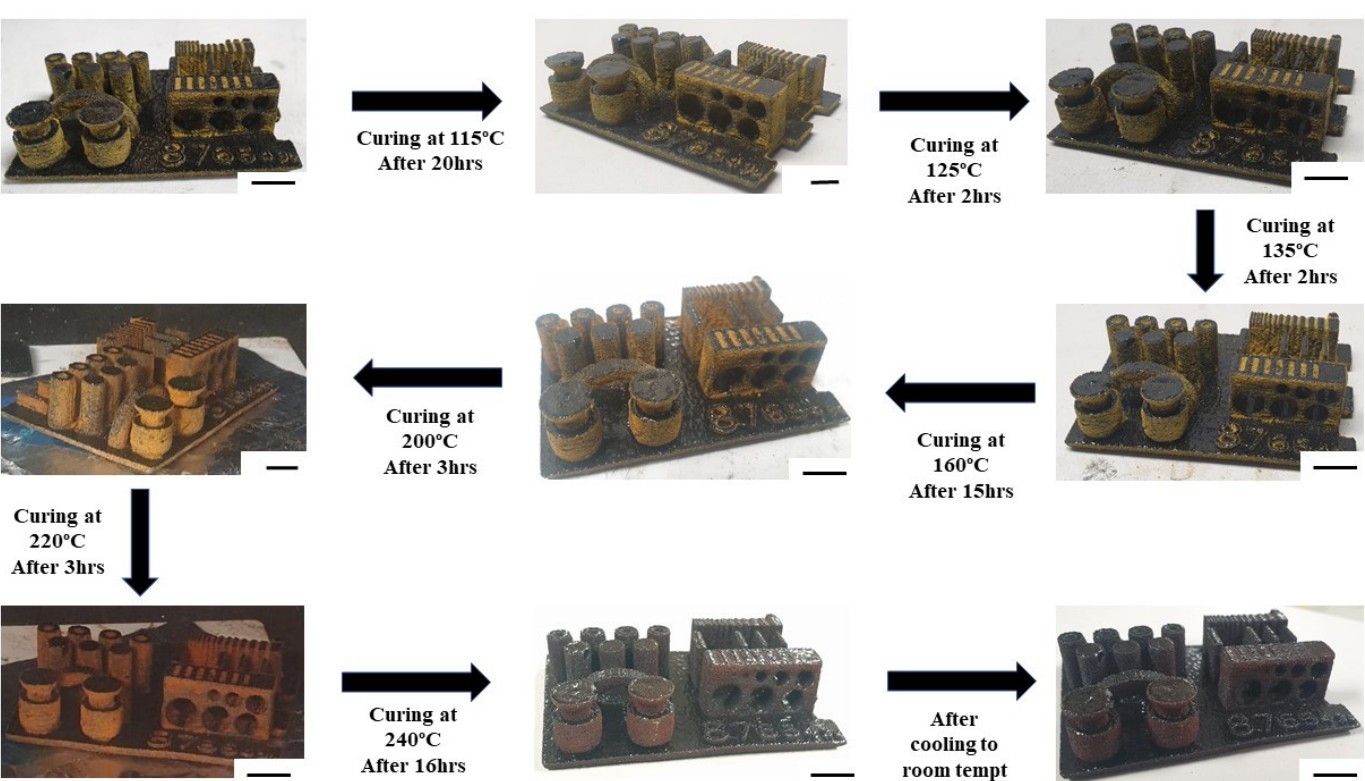

**Figure 1.** The step-by-step curing process of an SLS-printed complex sample (scale bar: 10 mm).

### 2.5. Characterization

2.5.1. Thermogravimetric Analysis

To determine the thermal degradation onset temperature of BMI powder, thermogravimetric analysis (TGA) was performed with a TGA 55 (TA instruments, New Castle, DE, USA). The temperature accuracy of the machine was $\pm 1$ °C, and the weighing accuracy was $\pm 0.01$%. The degradation onset temperature (DOT) was defined as the temperature at which 1 wt.% loss occurred. TGA scans were performed from 25 to 800 °C with a heating rate of 10 °C/min according to the ASTM standard E1131 [18]. The sample purge rate was 60 mL/min, and the balance purge rate was 40 mL/min under the Nitrogen flow of 20 psi. A sample in the weight range of 3 to 5 mg was placed in a high-temperature platinum pan and its mass loss was recorded by the TA Instrument's Trios software.

2.5.2. Differential Scanning Calorimetry

To establish the print bed temperature during the SLS 3D printing process of a powder, the melting point onset temperature should be known. To identify the melting onset temperature of BMI powder, DSC tests were performed. In addition, DSC tests were performed to measure the degree of curing of BMI/CMB (cured samples). A Discovery 250 DSC instrument (TA instrument, New Castle, DE, USA) was used, having the temperature accuracy and precision of the instrument of $\pm 0.1$ and $\pm 0.01$ °C, respectively. To avoid contamination due to heating and cooling sequences, the test samples were sealed using aluminum T-zero pans and lids. Initially, the temperature was equilibrated at 25 °C and then ramped at a rate of 5 °C/min, from 25 to 230 °C. To avoid sample degradation within the DSC test chamber, the DSC temperature was chosen based on TGA. Trios software was used for DSC analysis.

As the maximum operating temperature of DSC 250 (TA instrument, New Castle, DE, CA, USA) is 250 °C, another high-temperature DSC machine was used to measure the degree of curing of cured thermoset printed samples. For this purpose, Netzsch DSC 404 F1 Pegasus (Burlington, MA, USA) was used, which runs DSC with a wide temperature range within $-150$ to 2000 °C. Each sample of weight $22 \pm 6$ mg was placed into an alumina crucible (diameter: about 6 mm, volume: 85 μL) and heated at 5 K/min under a continuous 30 mL/min Argon gas flow (purity 99.999%, Airgas). The temperature and sensitivity of the DSC were calibrated by melting several metal standards (In, Bi, Sn, Zn, Al, Ag, and Au). Specifically, for each metal, the temperature was adjusted by measuring the melting point and comparing it with the expected value, while the sensitivity was calibrated by measuring the melting peak area (μV·s/mg) and adjusting with the enthalpy (J/g). DSC curves for BMI/CMB samples with different degrees of curing were analyzed. The area of the exothermic peak was calculated with the NETZSCH Proteus® software (Selb, Germany) with a linear baseline. The software calculated the area by integrating between two limits. From the areas of exothermic peaks, the degree of curing was calculated with the following equation [19]:

$$\text{Degree of Curing} = 1 - \frac{\Delta H_t}{\Delta H_{unc}} \tag{2}$$

where $\Delta H_t$ is the residual enthalpy of the reaction of a sample cured for time t, and $\Delta H_{unc}$ is the total enthalpy of the curing reaction for an uncured sample.

2.5.3. Scanning Electron Microscopy

Scanning Electron Microscopy (SEM) was performed to observe the BMI particle sizes, morphology, and the fusion of the powder during the SLS printing and curing process. The IT 500 LV Scanning Electron Microscope (JEOL USA, Inc., Pleasanton, CA, USA) was used to perform SEM. To obtain high-resolution images, gold sputtering was performed before performing SEM characterization. Creating a conductive layer of metal using gold on the sample through gold sputtering inhibits charging, reduces thermal damage, and improves the secondary electron signal that is required for topographic examination in the SEM. The 108 Auto/Se sputter coater (TED PELLA, Inc., Redding, CA, USA) was used to

perform gold sputtering. For gold sputtering, a 0.3 psi nitrogen flow was set, and 10 micron of gold-sputtered thickness was deposited on SEM samples.

### 2.5.4. Fourier Transform Infrared Spectroscopy

The Fourier Transform Infrared Spectroscopy (FTIR) of BMI/CMB printed specimens with different degrees of curing was recorded on a Fisher Scientific NICOLET iS5 Spectrometer (300 Industry Drive, Pittsburgh, PA, USA). This device was equipped with an iD7 ATR accessory (Thermo Fischer Scientific Inc., Waltham, MA, USA), and FTIR for each of the samples was performed by scanning the samples 16 times with 4 $cm^{-1}$ spectral resolution, and the curves were analyzed using OMNIC software version 8.0. Chemical reactions and crosslinking phenomena during curing processes were investigated through FTIR.

### 2.5.5. Dynamic Mechanical Analysis

The storage modulus (G′), loss modulus (G″), and damping factor (tanδ) of the fabricated and cured parts were determined by performing the Dynamic Mechanical Analysis (DMA). The ASTM D4065 [20] standard was followed for the dimension of the sample for DMA. The DMA 850 (TA instrument, New Castle, DE, USA) with modulus precision ±1% was used to perform DMA. To ensure firm gripping of samples by the DMA 850 machine's sample holder, the BMI/CMB DMA sample was printed 4 mm longer in length. A test specimen with dimensions of 60 × 13 × 3 mm was fabricated using the SLS 3D printer. DMA tests were performed from 30 to 320 °C at a heating rate of 3 °C/min with the test frequency of 1 Hz and 20 μm amplitude.

### 2.5.6. Compression Test

Compression testing was performed using an Instron 5969 dual-column testing system with a maximum capacity of 50 kN. The ASTM C365-00 [21] test standard was followed to perform the compression testing. The crosshead speed was 0.5 mm/min. For the flatwise compression test, CMB-blended BMI was fabricated as rectangular blocks at 25 × 25 × 12 mm.

The flatwise compression modulus, $E_c$, was calculated by the following equation [22]:

$$E_c = \frac{mt}{A} \tag{3}$$

where m is the slope of the initial region of the load-deflection curve, A is the cross-sectional area of the sample, and t is the thickness of the printed part. To statistically compare the test results, at least five tests were performed in each category.

## 3. Results and Discussion

### 3.1. BMI Thermoset Powder Morphology

Representative SEM images of BMI thermoset powders as-received and after sieving are shown in Figure 2. As-received samples showed that some particles were large (larger than 80 μm), while on the other hand, the sieved powder had a 16 to 53.3 μm particle size range.

### 3.2. Thermogravimetric Analysis

Thermogravimetric analysis (TGA) was carried out to analyze the thermal stability of the BMI thermoset. Figure 3 shows that the degradation of thermoset BMI powder occurred in two stages of weight loss (higher resolution images of figures may be found in the Supplementary Materials). The first stage that occurred in the range of 300 °C indicates the structural decomposition of BMI powder. The second stage started at around 700 °C, corresponding to the combustion of residuals. The degradation onset temperature (DOT) (defined as the 1% reduction of the weight) of the thermoset was 243 °C. Besides determining the DOT and thermal stability of BMI powder, a TGA graph was used to determine the maximum operating temperature for the DSC run of the powder.

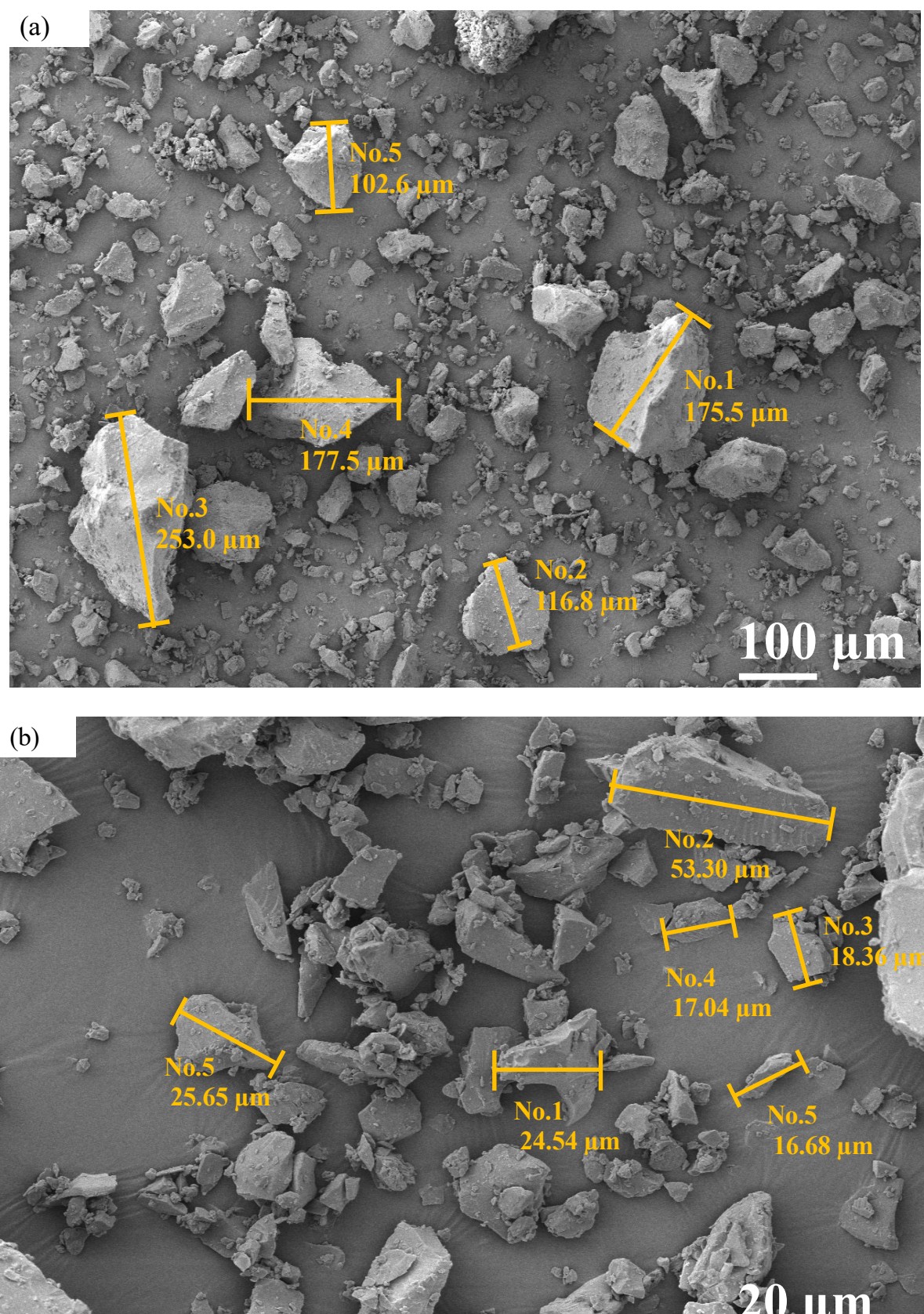

**Figure 2.** SEM of BMI thermoset powder: (**a**) before sieving and (**b**) after sieving.

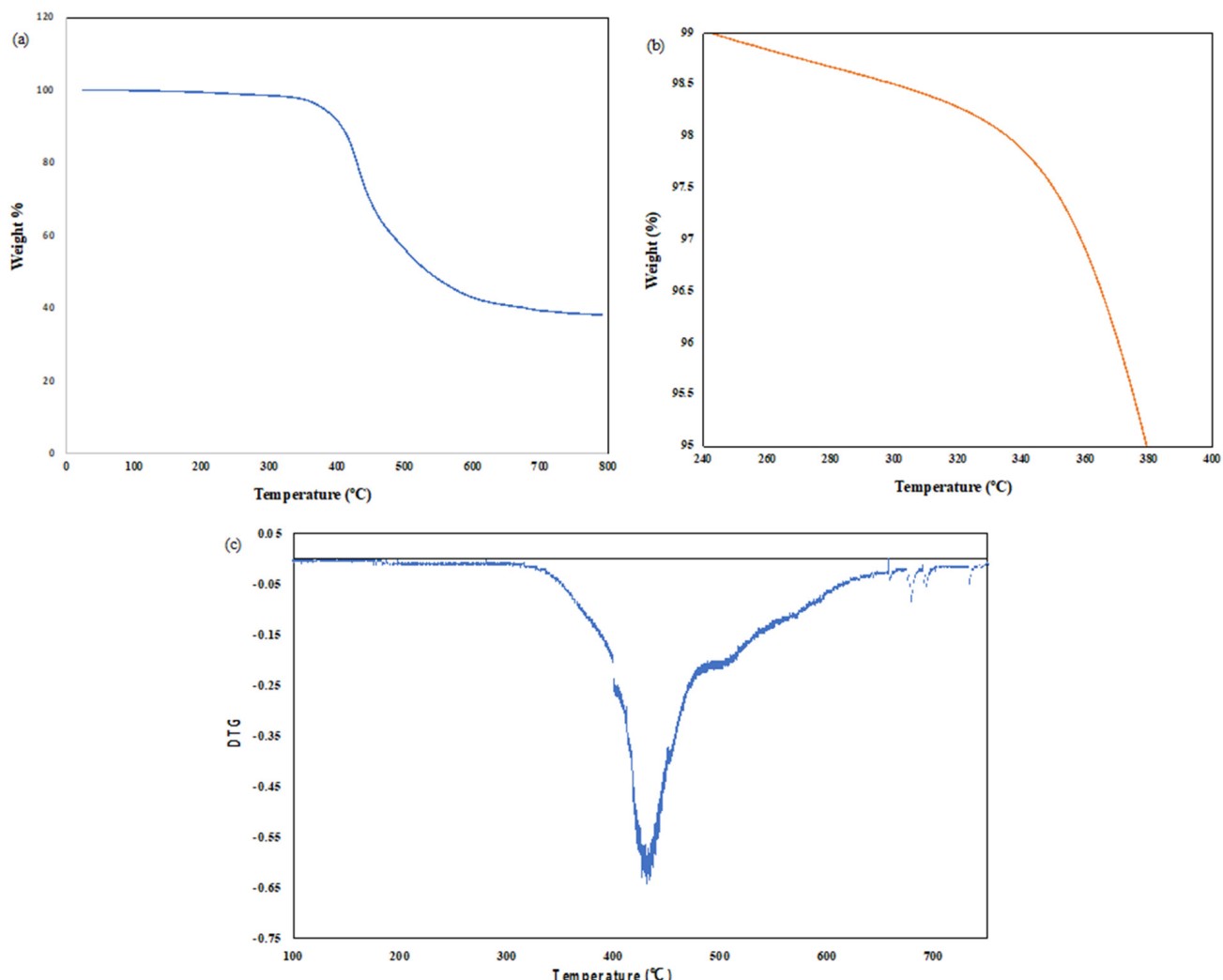

**Figure 3.** Representation of TGA of BMI thermoset: (**a**) weight loss thermogram and (**b**) 1% to 5% weight loss in thermogram. (**c**) Differential thermogravimetric properties of BMI thermoset.

### 3.3. Selective Laser Sintering and Curing of BMI/10 Vol% CMB

From the SEM (Figure 2), it was observed that the particles were not spherical in shape; therefore, to obtain uniform powder distribution during the SLS process and to improve powder flowability, a higher layer height (150 μm) was set in the software, and powder preheating at 70 °C was performed. To obtain the desired geometrical accuracy and quality of the printed part, different printing parameter sets were implicated, and the resultant specimen appearance is summarized in Table 2.

**Table 2.** Summary of SLS parameters and DMA specimen appearance.

| ES | MEP | CE | LH (mm) | HS (mm) | Specimen Appearance | Figure |
|----|-----|----|---------|---------|----------------------|--------|
| 10 | 400 | 0.5 | 0.125 | 0.36 | No shifting but curling occurred, strong but oversized printed part. | |

**Table 2.** *Cont.*

| ES | MEP | CE | LH (mm) | HS (mm) | Specimen Appearance | Figure |
|----|-----|----|---------|---------|---------------------|--------|
| 3 | 200 | 0.5 | 0.125 | 0.36 | Fragile, no shifting or curling occurred, dimensional increment. |  |
| 3 | 200 | 0.5 | 0.175 | 0.1 | Lower part was severely damaged, deformation occurred, dimensional increment due to high ES values. (Built in on-edge direction) |  |
| 3 | 200 | 0.5 | 0.125 | 0.5 | No layer shifting, slightly increased in thickness due to high ES value. |  |
| 3 | 200 | 0.5 | 0.125 | 0.36 | No layer shifting, slightly increased in thickness due to high ES value, more stable. |  |
| 2 | 200 | 0.5 | 0.175 | 0.2 | Lower portion was damaged, slightly dimensional increment due to higher ES values. (Built in-on edge direction) |  |
| 2 | 400 | 0.2 | 0.150 | 0.36 | No layer shifting, dimensional increment in thickness. |  |
| 1 | 800 | 0.2 | 0.150 | 0.36 | Oversized in thickness. |  |
| 1 | 700 | 0.8 | 0.150 | 0.36 | Stable with accurate dimensions, best surface finishing. |  |

Eventually, the printing parameter set was optimized (as shown in Table 1), and SLS-printed DMA and compression testing samples were obtained, which were dimensionally accurate, stable, and with no shifting after printing. Figure 4 shows DMA and compression testing samples after the curing process. Figure 5 represents that after curing, the SLS-printed DMA sample was increased by 0.033% in length and 0.05% in width, and thickness was decreased by 0.0167% (higher resolution images of figures may be found in the Supplementary Materials).

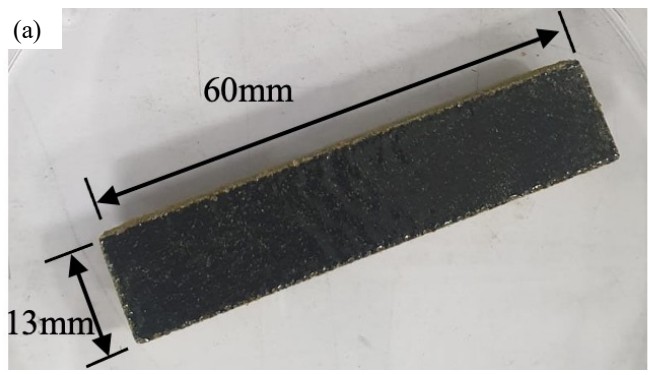
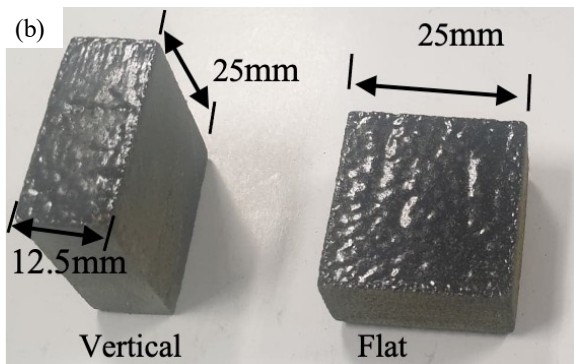

**Figure 4.** SLS-printed (**a**) DMA sample and (**b**) compression testing sample (printed at 2 different orientations).

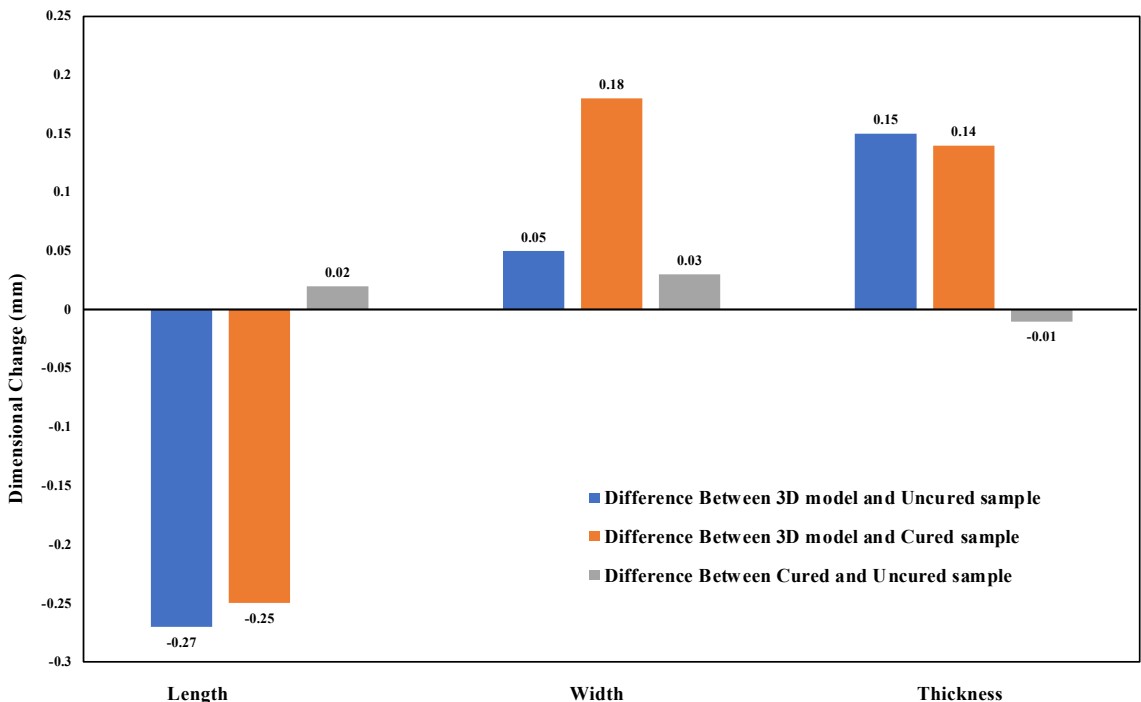

**Figure 5.** Dimensional changes of DMA samples in length, width, and thickness.

The complex 3D printer test sample was also printed and cured as shown in Figure 6. Initially, a single-step curing process was followed to cure printed specimens, but the complex shape could not hold its shape after curing at 200 °C (Figure 6a). Therefore, a step-by-step curing process (Figure 2) was implemented that helped to hold the shape and geometry of the complex 3D-printed test sample (Figure 6b) and KCNSC logo (Figure 6c).

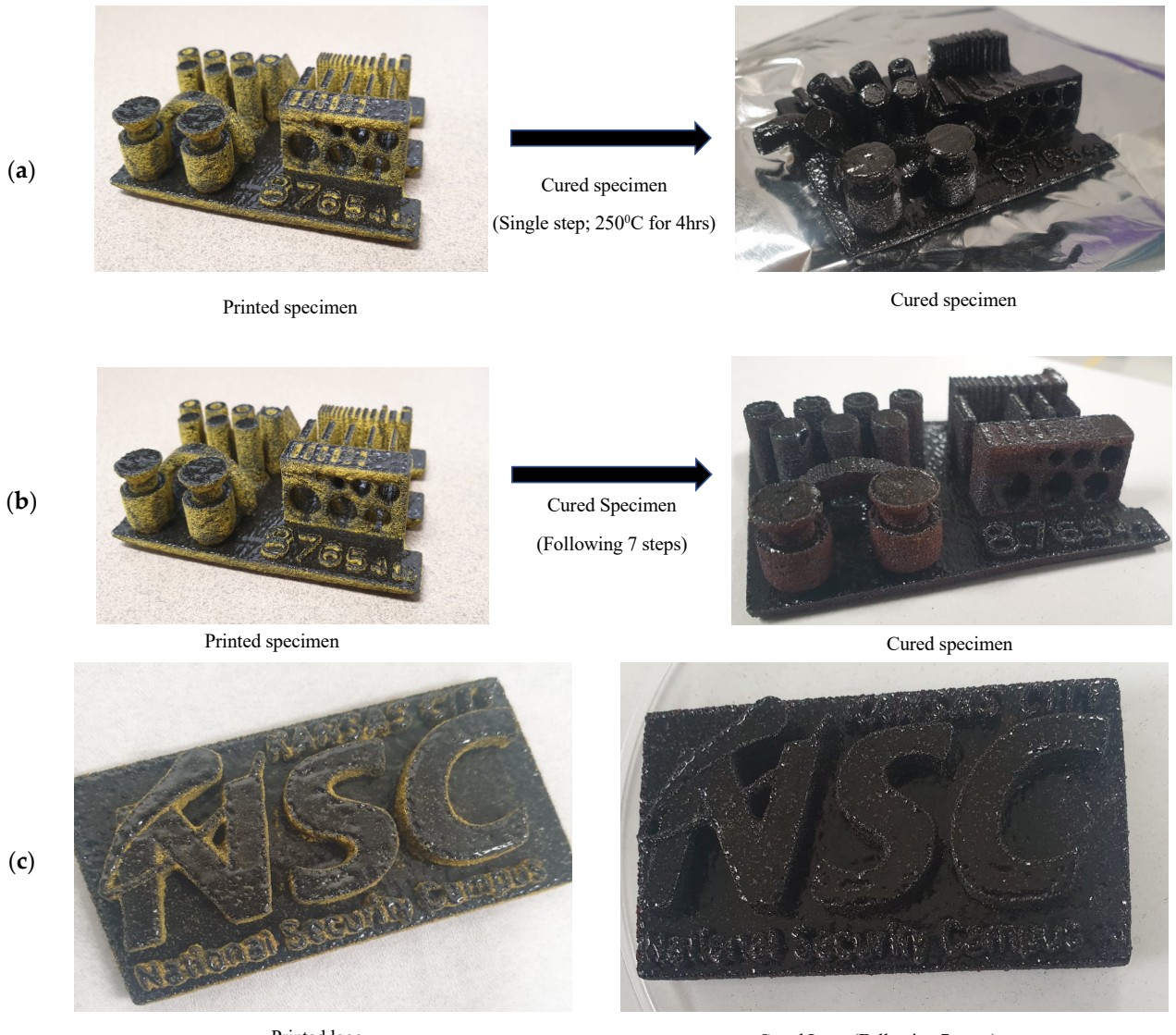

**Figure 6.** Curing of complex shape following (**a**) single stage (room temperature to 250 °C), (**b**) multiple steps, and (**c**) SLS-printed uncured and cured specimen for showcasing.

### 3.4. Differential Scanning Calorimetry

DSC tests were performed to figure out the melting point of BMI powder as well as to calculate the degree of curing of the printed part. As the DSC curve (Figure 7) shows an endothermic peak at 112.02 °C indicating the initial melting point of BMI, thus the bed temperature was set at 105 °C, slightly lower than the initial melting temperature [23] (higher resolution images of figures may be found in the Supplementary Materials).

In the DSC curve shown in Figure 8, the blue line shows an endothermic peak at 112.34 °C that indicates the glass transition temperature of the SLS-printed uncured sample. For partially cured samples (cured at 140 °C for 4 h and at 170 °C for 4 h), which are denoted by orange and black lines, endothermic peaks were shown at 136.36 and 172.39 °C consecutively, followed by exothermic peaks. On the other hand, the green line, which denotes a fully cured sample, does not show any endothermic peak. As the $T_g$ point shifts from left to right, this phenomenon implies that the glass transition temperature and thermal stability of samples are being improved with the improvement of the degree of curing.

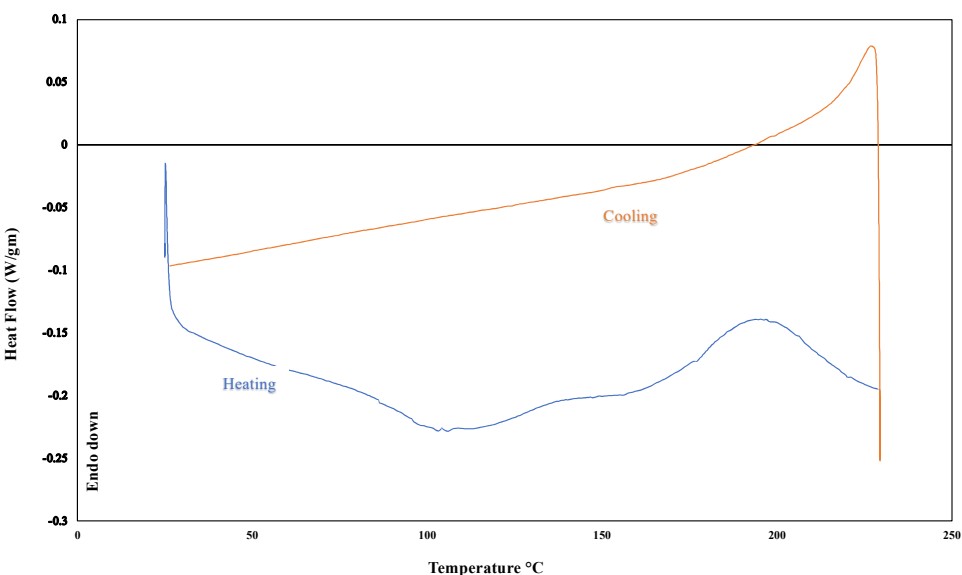

**Figure 7.** DSC curve of Bismaleimide resin powder.

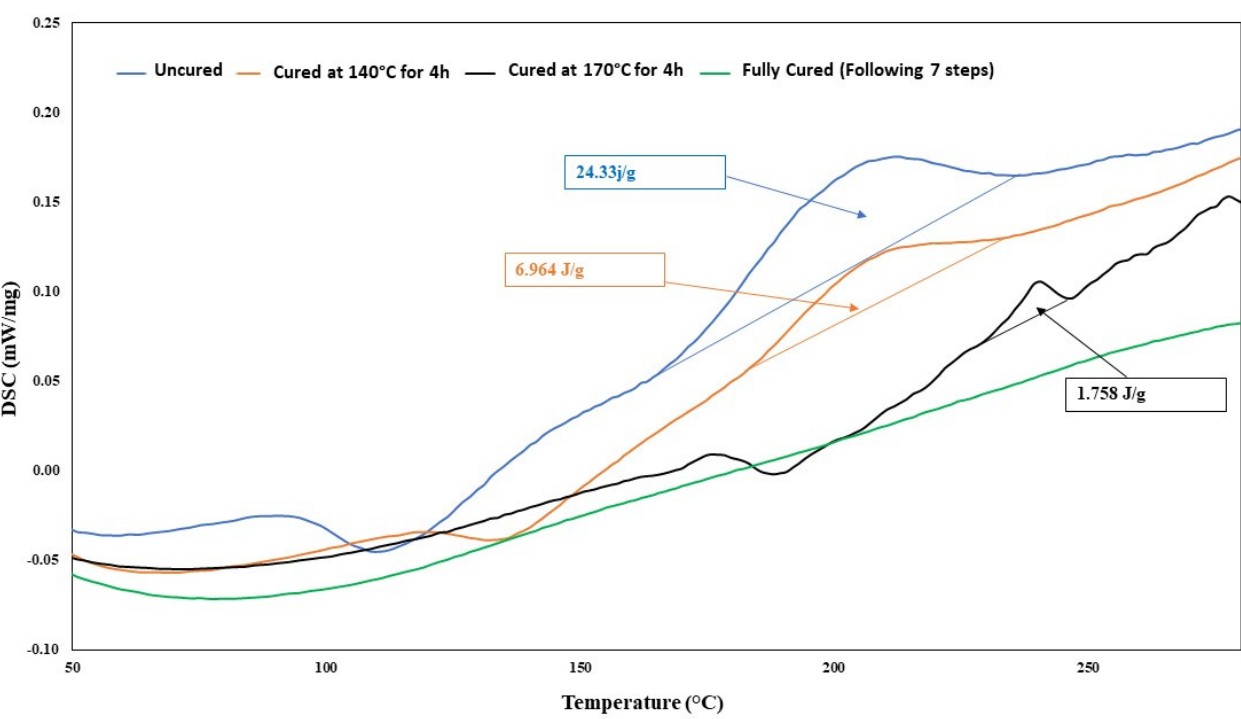

**Figure 8.** DSC curves of SLS-printed parts with different degrees of curing.

The exothermic region of the blue line is due to the curing reaction during the heating process and has a maximum temperature of 210.45 °C. The area under the exothermic peak, which is the total enthalpy of the curing reaction for an uncured sample ($\Delta H_{unc}$), is 24.33 J/g. For the partially cured samples, which are denoted by orange and black lines, the area under the exothermic peaks is 6.964 and 1.758 J/g, consecutively. The fully cured sample (green line) does not show any exothermic peaks. Therefore, as the molecular crosslinking proceeds during curing, the exothermic peak areas are decreased. The degree of curing was calculated by using Equation (2), and the bar chart (Figure 9) demonstrates how the degree of curing is being improved with the curing temperature and time.

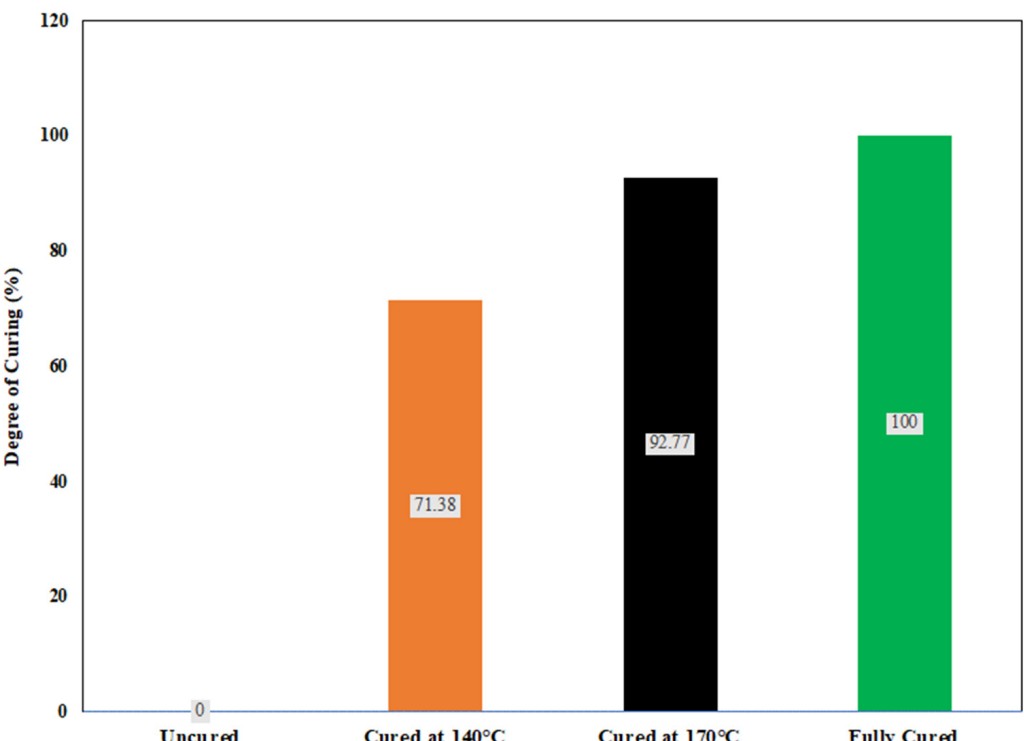

**Figure 9.** Improvement of the degree of curing with curing conditions.

### 3.5. Scanning Electron Microscopy

Figure 10 shows the SEM of the top, bottom, and cross-sectional view of the SLS-printed sample before and after curing. The bottom view (Figure 10b) represents that the powder was not uniformly melted, and the reason was that during the printing of the very first layer, some powders under the defined printing zone were attached to the printed part. The cross-section of the printed part represents (Figure 10c) that BMI was fully fused by laser energy. The SEM of the fully cured printed part represents fewer pores at the top surface (Figure 10a), and powder particles were uniformly melted and cured at the bottom surface. The circular hollow shapes at the cross-section of the cured sample indicate that the CMB fillers were uniformly embedded in the BMI matrix.

### 3.6. FTIR Spectroscopy

Curing of BMI/CMB printed samples at varying temperatures (at 4 h) results in different degrees of curing, which was investigated by DSC analysis. However, the curing mechanism needs to be further investigated. Figure 11 shows the FTIR spectra of BMI/CMB powder, uncured printed samples, and samples with different degrees of curing. The details of the uncured spectrum interpretation have been reported by other researchers [24]. During the curing process of BMI resin, homopolymerization occurs at a reactive double bond (H-C=C-H) in the maleimide groups through a free-radical addition reaction [25]. In this reaction, the maleimide group is converted into the succinimide group as a final product [26]. From FTIR spectroscopy (Figure 11), several changes have been observed after the polymerization through the curing process (higher resolution images of figures may be found in the Supplementary Materials). With the increase of the curing temperature, the disappearance of the peak at wavelength 1150 cm$^{-1}$ and the growth of a peak centered at 1185 cm$^{-1}$ are visible. The changes occur due to the conversion of the C-N-C maleimide group into the succinimide group, and these peaks strongly overlap. At 690 cm$^{-1}$, with the increasing degree of curing, the disappearance of the peak is the characteristic of maleimide groups (C=C-H).

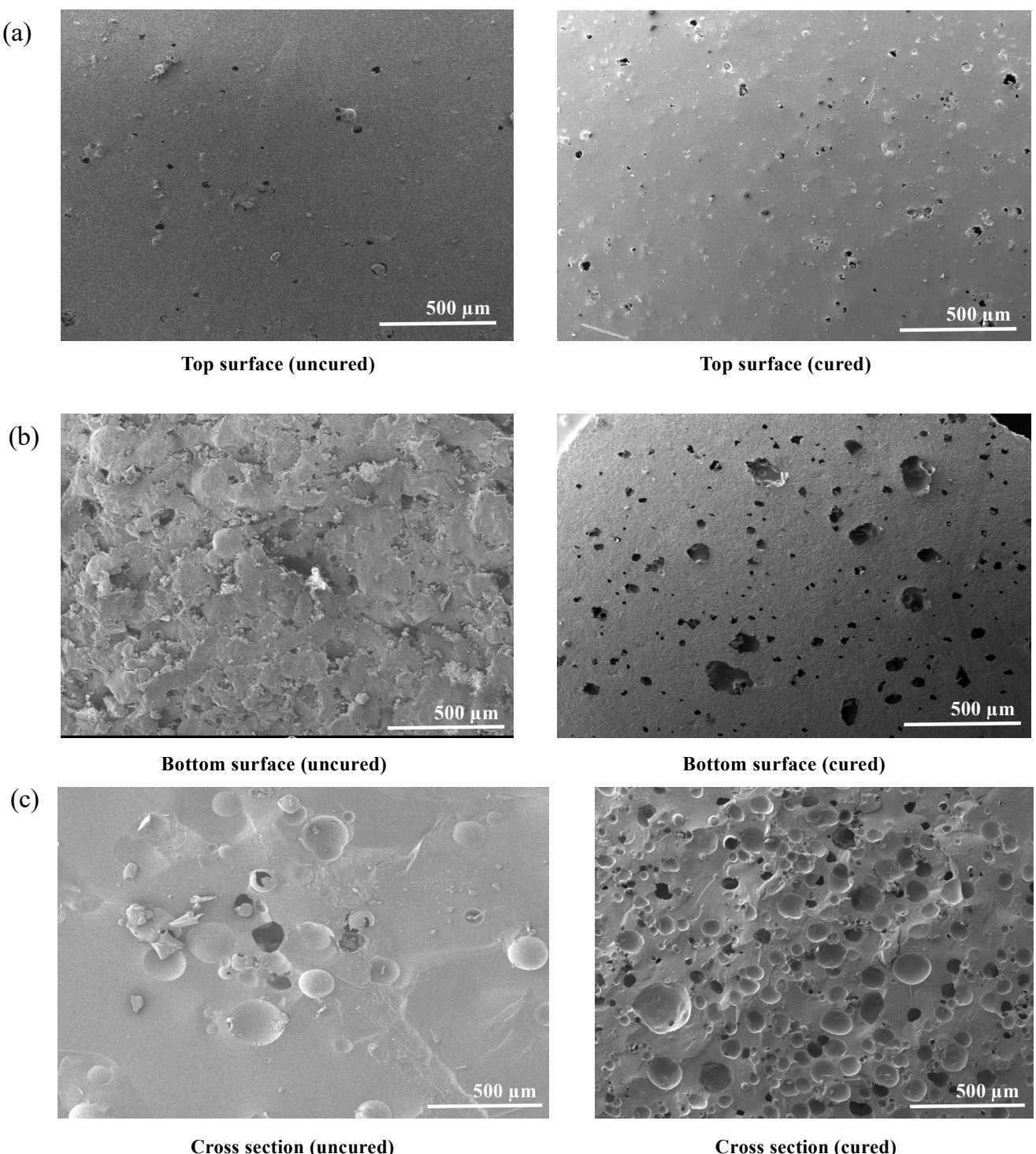

**Figure 10.** Scanning Electron Microscopy of SLS-printed uncured and cured specimens: (**a**) top surface, (**b**) bottom surface, and (**c**) cross-section.

After full curing (at 250 °C for 4 h), the FTIR spectroscopy shows that the peak at 1709 cm$^{-1}$ (maleimide C=O) is broadened as a result of overlapping of different peaks due to the formation of succinimide C=O. Several researchers have used the peaks at 1150 cm$^{-1}$ and 690 cm$^{-1}$ to analyze the curing reaction in BMI polymerization [24]. Figure 11 shows that chemical crosslinking and homopolymerization reactions are obvious with the progression of the degree of curing.

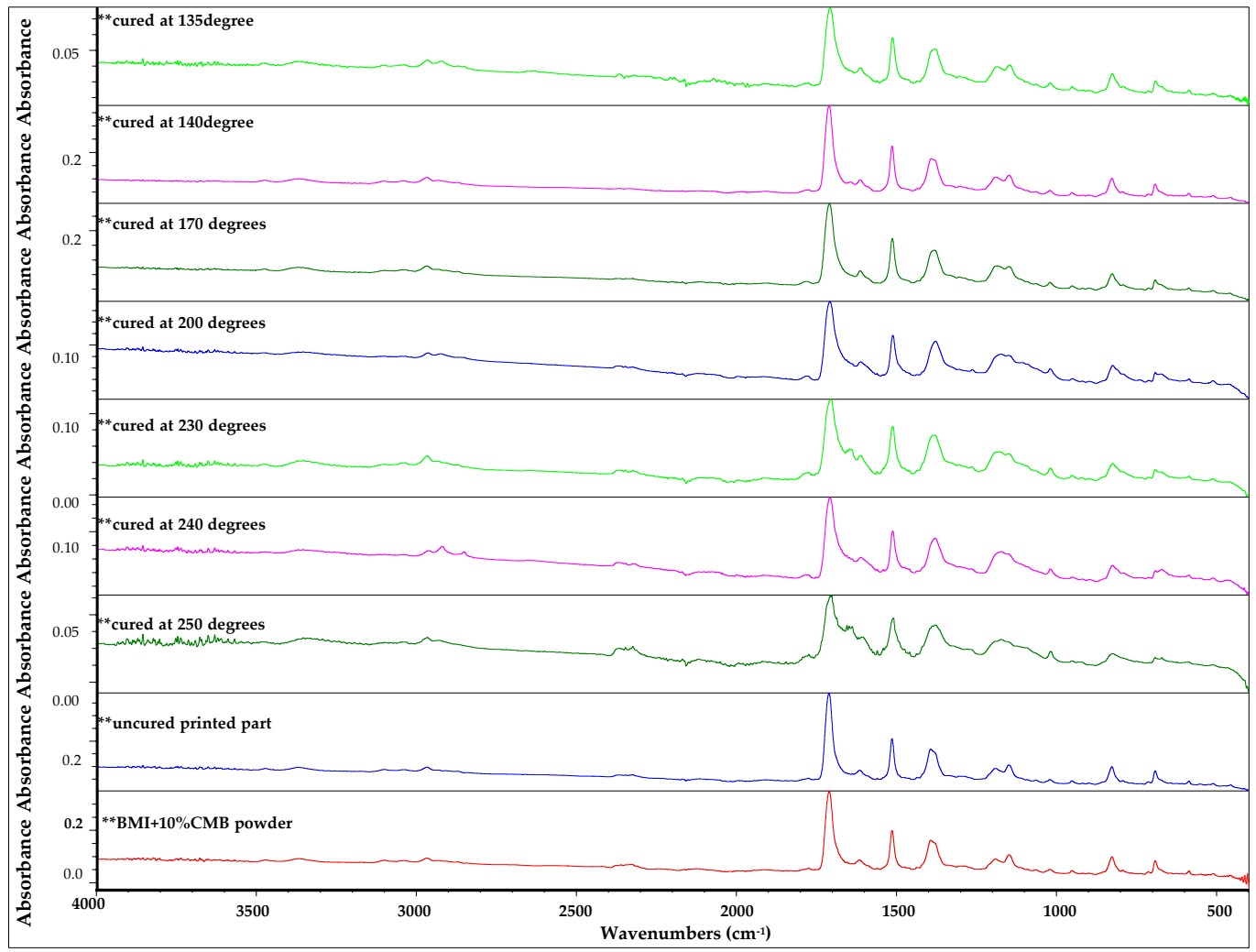

**Figure 11.** FTIR spectra of the BMI/CMB powder blend and the BMI/CMB printed part cured at different degrees of curing (for all cases, curing time was 4 h).

### 3.7. Dynamic Mechanical Analysis

Figure 12a shows the DMA for the uncured printed BMI/CMB sample. The downward trend of the storage modulus and the upward trend of the loss modulus after around 90 °C results in a sudden increment of the damping factor (tanδ) (loss modulus/storage modulus). The drastic increment of the damping factor indicates the rubbery behavior of printed uncured samples after 90 °C. As the printed part was not cured and the melting point of BMI was around 110–120 °C, running DMA testing of this sample at a higher temperature might damage the DMA analyzer. Thus, the tests were run at 115 °C. From the rising trend of the tanδ curve, it can be predicted that the glass transition ($T_g$) point would be obtained at around 120 °C [27]. Figure 12b represents the DMA of the printed part that has been cured by the following seven stages (Figure 1). Here, the damping factor (tanδ) peak is observed at 297.7 °C, which indicates the $T_g$ point for the fully cured sample. The plateau region of the storage modulus after around 300 °C indicates the crosslinking reaction inside the printed part [27]. Comparing Figure 12a and Figure 12b, the shifting of the tanδ peak from around 120 to 297.7 °C indicates the crosslinking and curing of the BMI/CMB printed part. Additionally, the DMA represents that the crosslinking and curing of the BMI/CMB printed part had made it thermally stable by increasing its $T_g$ point.

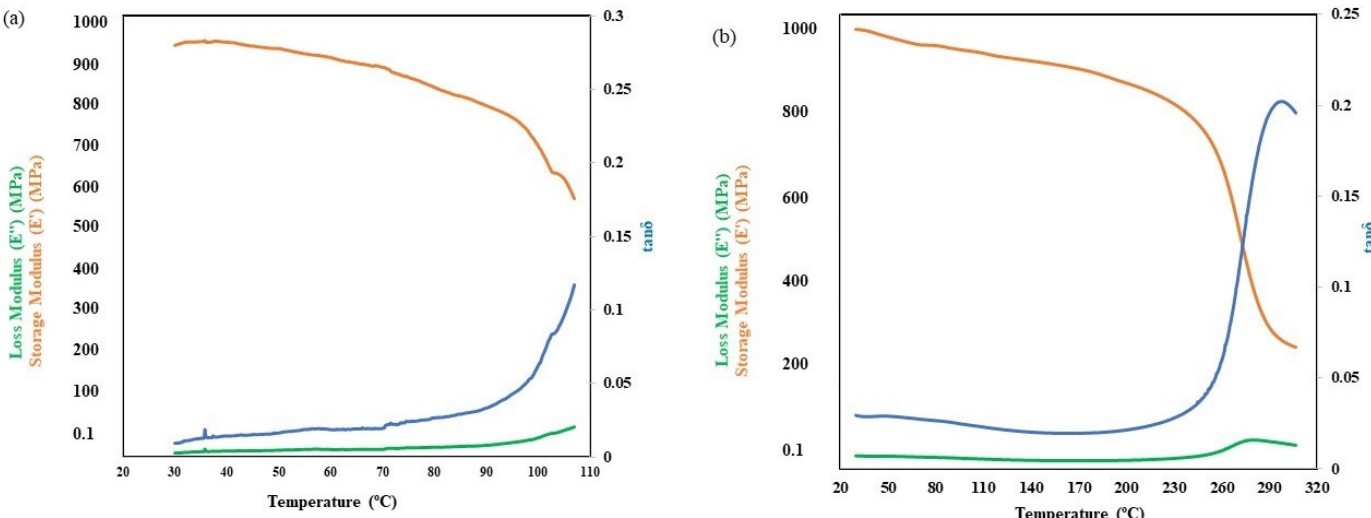

**Figure 12.** Representation of DMA results: (**a**) uncured printed sample and (**b**) cured printed sample.

### 3.8. Compression Test

The cured BMI/CMB blocks did not break with the compression testing machine's maximum load capacity, which was 50 kN. Therefore, the length, width, and height of the printed samples were reduced by 50%. The height of the specimen was chosen to make sure that the aspect ratio (height/width) was low so that the effect of the shear stress can be minimized [28]. Figure 13 represents the force–displacement curves generated by flatwise compression tests for uncured and cured printed BMI/CMB parts. Compression results from five samples from each type of specimen were investigated (Figures 13 and 14). It was observed that for uncured samples, up to a 0.98 kN load was carried out by the uncured printed part (Figure 13a). On the other hand, fully cured samples could carry up to 7.6 kN of load during the flatwise compression test (Figure 13b). A similar trend was observed in stress–strain curves (Figure 14, higher resolution images of figures may be found in the Supplementary Materials). Fully cured BMI/CMB printed parts showed up to 114.9 MPa stress before fracture (Figure 14b), while only up to 15.68 MPa ultimate stress was endured by uncured samples (Figure 14a). Additionally, it was found that the uncured samples were broken intermittently during compression tests due to a lack of molecular crosslinking that resulted in uneven curves. On the contrary, during the curing process of BMI/CMB printed parts, crosslinking through the formation of covalent bonds between individual chains of the polymer had made the BMI/CMB printed part stable. Therefore, the cured samples did not break before they reached their maximum load capacity, which resulted in smooth curves in both force–displacement and stress–strain graphs (Figures 13b and 14b).

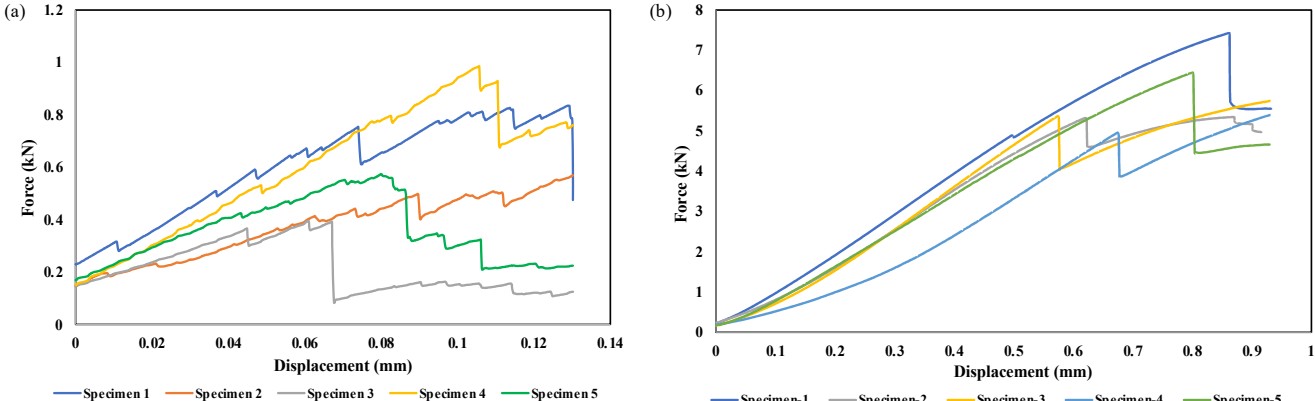

**Figure 13.** Force–displacement curves for (**a**) uncured and (**b**) cured samples.

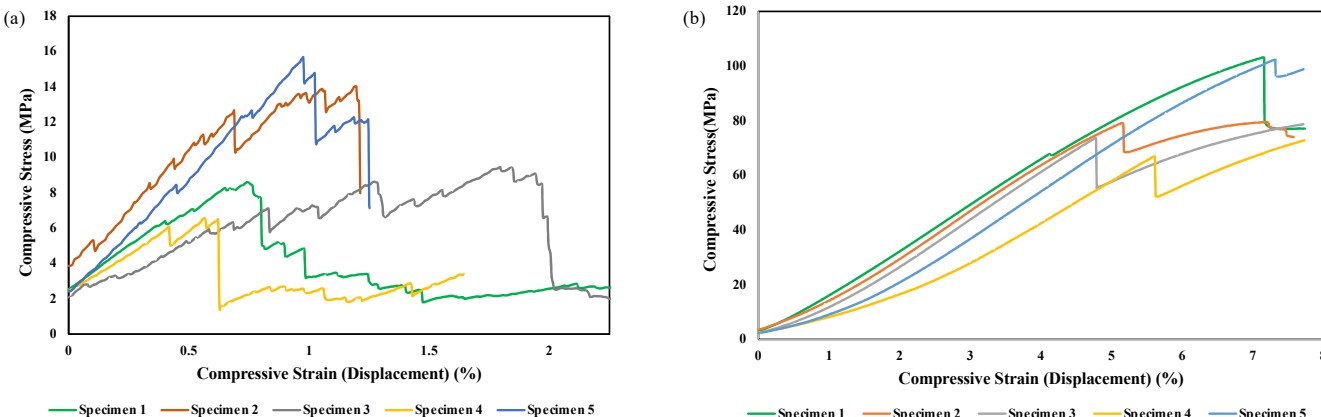

**Figure 14.** Compressive stress–strain curves for (**a**) uncured and (**b**) cured samples.

Using a 90% confidence interval and mean value of 5 samples, compressive strength and compression modulus bar charts of uncured and cured samples are represented in Figure 15a,b (higher resolution images of figures may be found in the Supplementary Materials). A significant difference in compressive strength between uncured and cured samples was observed (Figure 15a). The higher compressive strength of fully cured samples (87.23 MPa) than that of the uncured sample (9.664 MPa) indicates that the BMI/CMB printed parts were more stable after full curing. Similarly, the bar chart of compression modulus shows that cured BMI/CMB samples showed an 84.86% higher compression modulus than the compression modulus of uncured BMI/CMB samples. The higher compression modulus of fully cured BMI/CMB samples specifies higher stiffness than the uncured printed compression testing specimens.

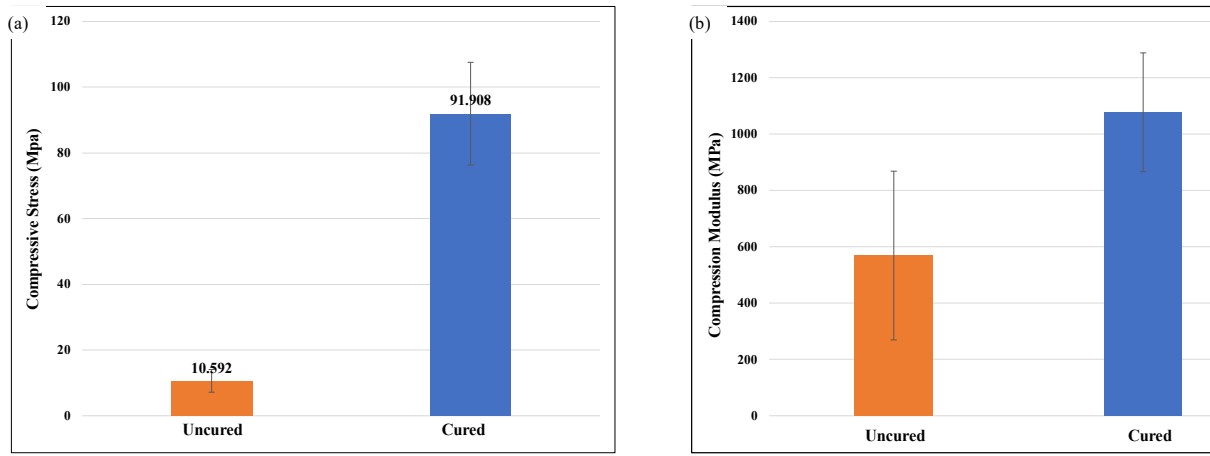

**Figure 15.** (**a**) Compressive strength and (**b**) compression modulus of the BMI/CMB printed part.

## 4. Conclusions

In this study, a thermoset Bismaleimide (BMI) powder was 3D printed using SLS to prove its printability, processability, and thermal stability after curing. A carbon microsphere filler material was added to the BMI powder matrix to achieve proper fusion of particles during the sintering process and better energy absorption. Additionally, optimized printing parameters were used, aiming to enhance dimensional accuracy, density, and surface finishing, followed by a step-by-step curing schedule to achieve full crosslinking in the composite material. The main goal of this research was to demonstrate high thermal stability at high service temperature and improved mechanical properties of the fully cured SLS printed parts. Indeed, the results portrayed a successful outcome of the 3D-printed material followed by curing, mainly that the increased value of $T_g$ temperature showcased

thermal stability after crosslinking or curing of the green parts, and also demonstrated zero-dimensional change and the ability to hold its shape. Correspondingly, the mechanical tests showed a significant distinction between uncured and cured samples: a higher compressive strength, compression modulus, as well as stiffness were achieved in cured samples due to proper crosslinking of BMI/CMB particles. Additionally, SEM images of the sintered parts showed a good interaction between the polymer matrix and the filler material that were homogeneously blended. For future work, the addition of CMB will be used to increase absorption efficiency. Even though the filler concentration was minimum in this research, the laser type is a key factor when sintering these additive particles into the matrix, such as the laser diode that works best with darker materials during the sintering process. Therefore, a different laser type (such as $CO_2$ laser) will be needed for 3D printing translucent thermoset materials by SLS. Nonetheless, the addition of filler material will not only impact the processability of the printed parts but also enhance the mechanical properties of the material and potentially reduce the density to achieve lightweight composites without affecting their integrity.

**Supplementary Materials:** The following supporting information can be downloaded at: https://www.mdpi.com/article/10.3390/jcs6020041/s1, high resolution images of Figures 1–15.

**Author Contributions:** Conceptualization, M.S.H., J.W., T.R. and Y.L.; methodology, M.S.H., J.K., Y.L., S.E.H., S.S., S.C., C.M., K.M.M.B. and J.E.R.; software, M.S.H.; validation, M.S.H., Y.L., K.M.M.B. and S.E.H.; formal analysis, M.S.H.; investigation, M.S.H., J.K. and Y.L.; resources, Y.L.; data curation, M.S.H., E.S., S.S. and S.E.H.; writing—original draft preparation, M.S.H. and S.S.; writing—review and editing, M.S.H., S.S. and K.M.M.B.; visualization, M.S.H. and Y.L.; supervision, Y.L.; project administration, Y.L. and J.W.; funding acquisition, Y.L. and J.W. All authors have read and agreed to the published version of the manuscript.

**Funding:** This research was funded by [NNSA. DE-NA 0003865] grant number [DE-NA-0004051].

**Informed Consent Statement:** Not applicable.

**Conflicts of Interest:** The authors declare no conflict of interest.

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
