# Peer review of "Selective Laser Sintering of High-Temperature Thermoset Polymer"

_jcs, doi:10.3390/jcs6020041_

Round 1

Reviewer 1 Report

The authors printed a thermoset resin, BMI, using SLS, and investigated the thermal stability of printed parts after a multi-step post-curing. Furthermore, they studied the crosslinking mechanism during the printing and curing process. Accordingly, they successfully optimized the SLS printing parameters for BMI thermoset powder to obtain dimensionally accurate printed parts; moreover, they found the best curing/crosslinking method, the multiple temperature steps, or the step-by-step curing process. The topic of the paper is interesting, and it was written well. The performed tests are high enough to investigate the printing process and the printed parts. The selected geometry for SLS printing was complex enough to support the excellent printability of the BMI resin. However, there is the main concern about the title of the paper. As the authors provided in the text, CMB was used to increase the laser absorbability of BMI. Thus, BMI/CMB may not be considered as a composite. On the other hand, if it may be considered a composite, CMB's effect on different properties of the composites must be investigated. Altogether I could recommend the paper for publication in the Journal of Composite Science.

Minor comment.

C1. The texts of Figures 1 and 11 are not clear. They should be reconsidered

Author Response

Thank you for your suggestion on paper title and improvement of images!

The title of the paper has been changed, and the new title is “Selective Laser Sintering of High-Temperature Thermoset Polymer”

The texts of the figure 1 and 11 have been made clearer by increasing text sizes and converting the texts in bold form where necessary. You can find the updated figures in the new manuscript.

Reviewer 2 Report

This work is novel in SLS. Thermoset with two curing stages is used as feedstock materials here. In conventionally, only thermoplastic is available in SLS. It proposed a new approach for composite development in 3D printing. I suggested accepting this work with minor revision. 

Please improve the quality of images using professional software to edit. 

Author Response

That is a very good suggestion on quality of images. Thank you!

Some pictures have been upgraded to make them clearer. Also, we are uploading all the images separately so that editorial office can edit them using professional software if needed. If any more data is needed, we will be happy to provide.